# Universal perpendicular orientation of block copolymer microdomains using a filtered plasma

Jinwoo Oh [1,2], Hyo Seon Suh[3], Youngpyo Ko[1], Yoonseo Nah[1], Jong-Chan Lee[2], Bongjun Yeom[4], Kookheon Char[2], Caroline A. Ross [5] & Jeong Gon Son [1,6]

Sub-10 nm patterns prepared by directed self-assembly (DSA) of block copolymer (BCP) thin films offer a breakthrough method to overcome the limitations of photolithography. Perpendicular orientation of the BCP nanostructures is essential for lithographic applications, but dissimilar surface/interfacial energies of two blocks generally favour parallel orientations, so that the perpendicular orientation could only be obtained under very limited conditions. Here, we introduce a generalized method for creating perpendicular orientations by filtered plasma treatment of the BCP films. By cross-linking the surface of disordered BCP films using only physical collisions of neutral species without ion bombardment or UV irradiation, neutral layers consistent with the BCP volume fraction are produced that promote the perpendicular orientations. This method works with BCPs of various types, volume fractions, and molecular weights individually at the top and bottom interfaces, so it was applied to orientation-controlled 3D multilayer structures and DSA processes for sub-10 nm line-spacing patterns.

[1] Photo-Electronic Hybrids Research Center, Korea Institute of Science and Technology, Seoul 02792, Republic of Korea. [2] School of Chemical & Biological Engineering, Seoul National University, Seoul 08826, Republic of Korea. [3] imec, Kapeldreef 75, 3001 Leuven, Belgium. [4] Department of Chemical Engineering, Hanyang University, Seoul 04763, Republic of Korea. [5] Department of Materials Science and Engineering, Massachusetts Institute of Technology, Cambridge, MA 02139, USA. [6] Division of Energy & Environment Technology, KIST School, Korea University of Science and Technology, Seoul 02792, Republic of Korea. Correspondence and requests for materials should be addressed to J.G.S. (email: jgson@kist.re.kr)

The semiconductor industry is contemplating a variety of ways to go beyond the resolution limit of conventional photolithography[1] in order to create sub-10 nm patterns that can increase microprocessor density and improve performance. Among these methods, directed self-assembly (DSA) of diblock copolymers (BCPs) has been studied extensively[2–5]. In order to use these spontaneously-formed patterns as a lithographic template, perpendicular orientation of the microdomains in thin film geometry is desirable to simplify pattern transfer[6,7]. However, the difference in surface/interface energy that typically exists between the two blocks of the BCP instead promotes in-plane microdomain orientation[8]. Great efforts have been made to realize a perpendicular orientation through various methods[9–18]. Perpendicular orientation has been implemented using external forces such as an electric field[10] or a solvent concentration gradient during solvent annealing[15–17], and by using random copolymer brushes[11–14,19] or surfactants[18] to provide neutral conditions at the surface and interface of the film. However, these methods are limited to specific BCPs and the conditions required for neutralization are narrow and vary with the BCP volume fraction[13], so it is necessary to separately tune the conditions for each block copolymer (BCP).

To achieve directionally controlled and complex BCP patterns by DSA, pre-patterned templates are required to guide the self-assembly[5]. Among them, graphoepitaxy is based on a topographical template[20–22] and chemoepitaxy[23–26] is based on a chemical pattern on the substrate. These techniques can be combined with neutralization of the surface of the film to minimize defects in the self-assembled pattern[27], e.g., by introducing a neutral top coat[16,28]. The BCP microdomains can be disturbed during the formation and removal of the top coat, and the neutral condition of the top coat needs to be tailored to the specific BCP[29]. Recently, Suh et al.[30] introduced a chemically cross-linked top coat layer formed by initiated chemical vapor deposition, although implementation of this method is complicated, whereas Lu et al. plasma-oxidized a polymer surface to obtain a neutral top surface[31]. Ar plasma treatment is a simple method to cross-link polymer surfaces[32,33] which can provide a solution to these challenges and which is compatible with planar processing. However, the combination of accelerated ion bombardment and vacuum ultraviolet/ultraviolet (VUV/UV) irradiation in a plasma both etches and cross-links a polymer[34–37], which leads to damage and thus is not suitable for use as a neutral layer for defect-free DSA patterning.

Here, we introduce a filtered plasma treatment to selectively block the VUV/UV irradiation and ion bombardment from reaching the BCP film while allowing physical collisions by neutral plasma species. This process effectively cross-links the BCP chains at the film surface to form an inherently neutral surface without etching damage or modification of the block volume fraction. This method can even be applied to UV-sensitive methacrylate[34,37] BCP films. We show that this simple process enables the perpendicular orientation of various BCPs including polystyrene-*block*-poly(methyl methacrylate) (PS-*b*-PMMA), polystyrene-*block*-polydimethylsiloxane (PS-*b*-PDMS), poly(2-vinylpyridine)-*block*-polystyrene-*block*-poly(2-vinylpyridine) (P2VP-*b*-PS-*b*-P2VP), polystyrene-*block*-poly(2-vinylpyridine) (PS-*b*-P2VP) and poly(methyl methacrylate)-*block*-polydimethylsiloxane (PMMA-*b*-PDMS) regardless of volume fractions, morphology, and annealing process.

## Results

### Plasma treatment of BCP film with the filter.
In this work we chose an argon plasma to minimize chemical reactions between the plasma and the film. During the plasma process, accelerated Ar ions

physically bombard the surface, forming a modified amorphous carbon-containing layer with a thickness of a few nanometers[38]. In addition, VUV/UV photons (originating from Ar emission at 104.8 and 106.7 nm and other emissions from residual gases) generated in the plasma cause a chemical modification of the film down to a depth of several hundred nanometers[37,39], and neutral species (i.e., metastable species and free radicals) also physically strike and modify the film surface.

A staggered bilayer filter consisting of two absorbers placed 200 μm apart with periodic 800 μm slits was introduced to block the light and ions travelling at near-normal incidence while allowing physical collisions by neutralized particles at oblique incidence[40,41], as shown in Fig. 1a, b. (The ions are neutralized by collisions with the filter[42].) This physical bombardment of the film results in cross-linking of polymer chains only near the surface without chemical changes from the VUV/UV irradiation or carbonization from accelerated ion bombardment[38]. The geometry of the staggered bilayer filter is described in Supplementary Fig. 1. Several articles have confirmed that these filter approaches can selectively block VUV/UV by at least 95%[34,35,40–42], enabling studies of the effect of the plasma on the polymer film surface[34,35,42] or as an ultra-low-damage surface treatment technique[40,41].

In the present work, the cross-linked layer at the surface formed by the physical collisions of neutral species retains the original volume fraction of the blocks and therefore can act as a neutral layer to the underlying BCP or to another layer of BCP applied on top of it. We demonstrate that the filtered plasma treatment produces a universal neutral top surface layer by spin-coating a BCP solution to form a disordered film, carrying out an Ar plasma treatment with the filter, then annealing the film to produce perpendicular orientation of the BCP microdomains at the top surface. To confirm the generality of our method, we successfully applied it to 200 nm thick films (sufficiently thick to minimize the effect of the bottom interface) of various BCPs with different morphologies, PS-*b*-PMMA (SML100, 50 kg mol$^{-1}$ PS-48 kg mol$^{-1}$ PMMA, $\phi_{PS}$ ~ 0.54, forming lamellae), PS-*b*-PDMS, (SDL43, 22 kg mol$^{-1}$ PS-21 kg mol$^{-1}$ PDMS, $\phi_{PS}$ ~ 0.49, forming lamellae and SDC16, 11 kg mol$^{-1}$ PS-5 kg mol$^{-1}$ PDMS, $\phi_{PS}$ ~ 0.67, forming PDMS cylinders in PS), PS-*b*-P2VP (SVL84, 40 kg mol$^{-1}$ PS-44 kg mol$^{-1}$ P2VP, $\phi_{PS}$ ~ 0.50, forming lamellae), P2VP-*b*-PS-*b*-P2VP (VSVL47, 12 kg mol$^{-1}$-23 kg mol$^{-1}$-12 kg mol$^{-1}$, $\phi_{PS}$ ~ 0.51, forming lamellae) and PMMA-*b*-PDMS (MDL22, 14 kg mol$^{-1}$ PMMA-8 kg mol$^{-1}$ PDMS, $\phi_{PMMA}$ ~ 0.59, forming lamellae) on pristine Si wafers.

Figure 2 shows the morphology at the surface after annealing of the BCP films, comparing samples treated with the filtered plasma with untreated samples. In the case of thermally annealed PS-*b*-PMMA, PS-*b*-PDMS, P2VP-*b*-PS-*b*-P2VP and PMMA-*b*-PDMS with lamellar microdomains shown in Fig. 2a–d, perpendicular microdomain orientations were formed at the surface by the filtered plasma treatment, but without the plasma treatment, terraced structures resulting from in-plane lamellae were formed. Particularly in the case of PS-*b*-PDMS, it is known that the difference in surface energy between PS (40.8 mN m$^{-1}$) and PDMS (20.4 mN m$^{-1}$) leads to PDMS at the surface which promotes in-plane oriented microdomains, but despite the large surface energy difference our method neutralizes the surface to promote a perpendicular microdomain orientation. A lower molecular weight asymmetric PS-*b*-PDMS (SDC16 in Fig. 2e) also produced perpendicular orientation of cylinders by thermal annealing after filtered plasma treatment. The filtered plasma process also led to a perpendicular orientation after solvent annealing, shown by the results of acetone-treated SDC16 film in Fig. 2f and chloroform-treated PS-*b*-P2VP (SVL84) film in Fig. 2g.

The thickness of the cross-linked layer from the filtered plasma was obtained from the morphology of SML100 films after

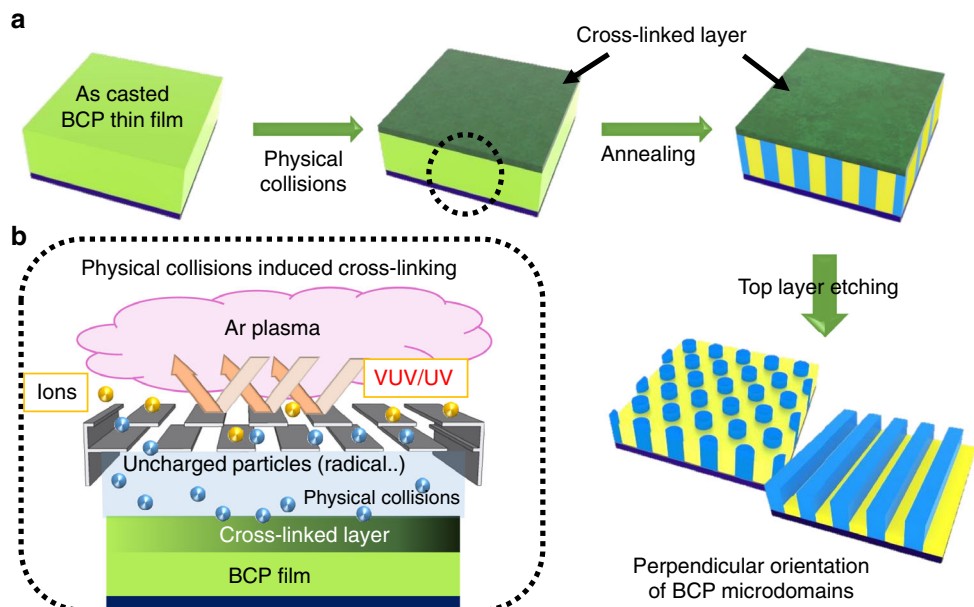

**Fig. 1** Filtered plasma treatment to form the cross-linked layer for perpendicular orientation of BCP microdomains. Schematics of **a** the plasma assisted process and **b** the filter which blocks ion bombardment and VUV/UV generated by the argon plasma. Only the Ar neutral particles impinge obliquely on the surface of the film, forming a thin cross-linked layer without composition changes or etching. The cross-linked layer exhibits a neutral surface affinity for the BCPs, so that the perpendicular orientation of the BCP microdomains in the film after annealing can be induced regardless of the type, molecular weight, or fraction of the BCP

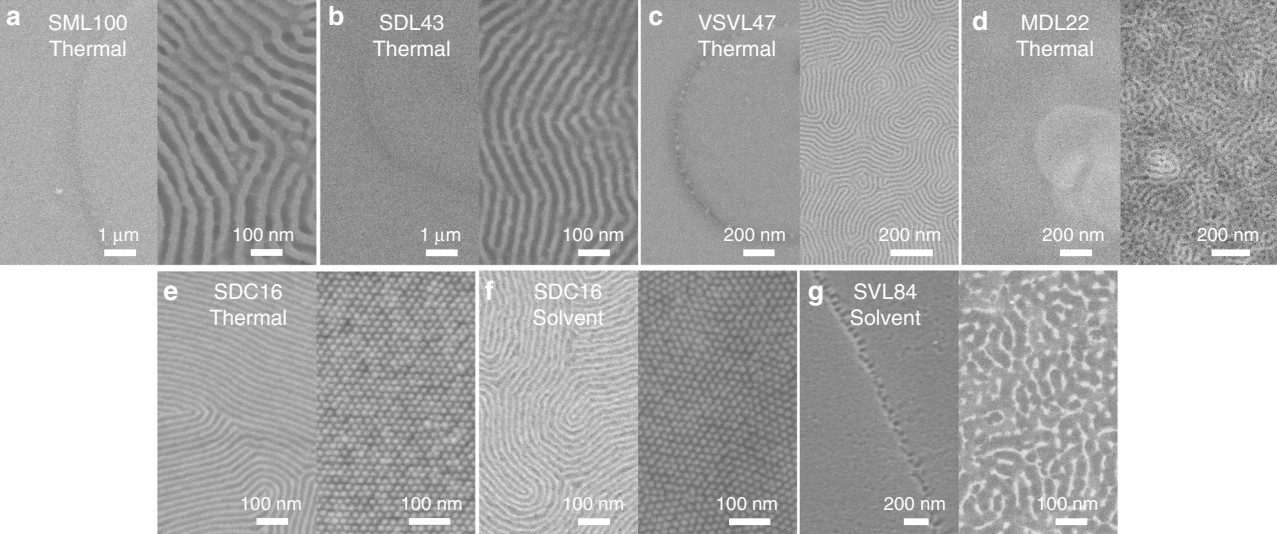

**Fig. 2** Morphology differences of various annealed BCP films with and without the filtered plasma treatment. Top view SEM images of thermally annealed PS-*b*-PMMA (**a**, SML100), lamellar PS-*b*-PDMS (**b**, SDL43), P2VP-*b*-PS-*b*-P2VP (**c**, VSVL47), PMMA-*b*-PDMS (**d**, MDL22) and cylinder PS-*b*-PDMS (**e**, SDC16), and solvent annealed (**f**) SDC16 and (**g**) PS-*b*-P2VP (SVL84) without (left) and with (right) the filtered plasma treatment. The annealed BCP films without the plasma treatment showed (**a**–**d** and **g**, left) a terraced structure from incommensurate thickness of the parallel lamellae and (**e**, **f**, left) parallel cylindrical microdomains. On the other hand, perpendicular orientation of lamellae (**a**–**d** and **g**, right) and cylinders (**e**, **f**, right) were obtained on the surface of various BCP films with filtered plasma treatment

annealing (Supplementary Fig. 2). The unetched as-cast film, the plasma-treated BCP film (200 W, 10 s with the filter) and the plasma-treated film after thermal annealing at 220 °C for 4 h all showed a poorly ordered morphology at the top surface. However, oxygen etching (O₂ RIE at 90 W, 10 mTorr) of the annealed film removed the cross-linked surface layer to reveal perpendicular lamellae. An etch of 5 s or more (etch depth ~4 nm measured by spectral reflectometry) was sufficient to expose the lamellae, but a 2 s etch (depth ~ 1.5 nm) was not, implying that the thickness of the cross-linked layer is between 1.5 and 4 nm.

Our result is in good agreement with other studies that the plasma treatment of polymer films results in a 1.5–2 nm thick cross-linked layer with higher refractive index[34] and modulus[38].

**Structural characterizations of BCP films with filtered plasma at different positions.** The filtered plasma treated cross-linked layer can act as a neutral interface not only at the top surface but also at the substrate interface similar to a random copolymer brush (Supplementary Fig. 3). This was demonstrated by spin-

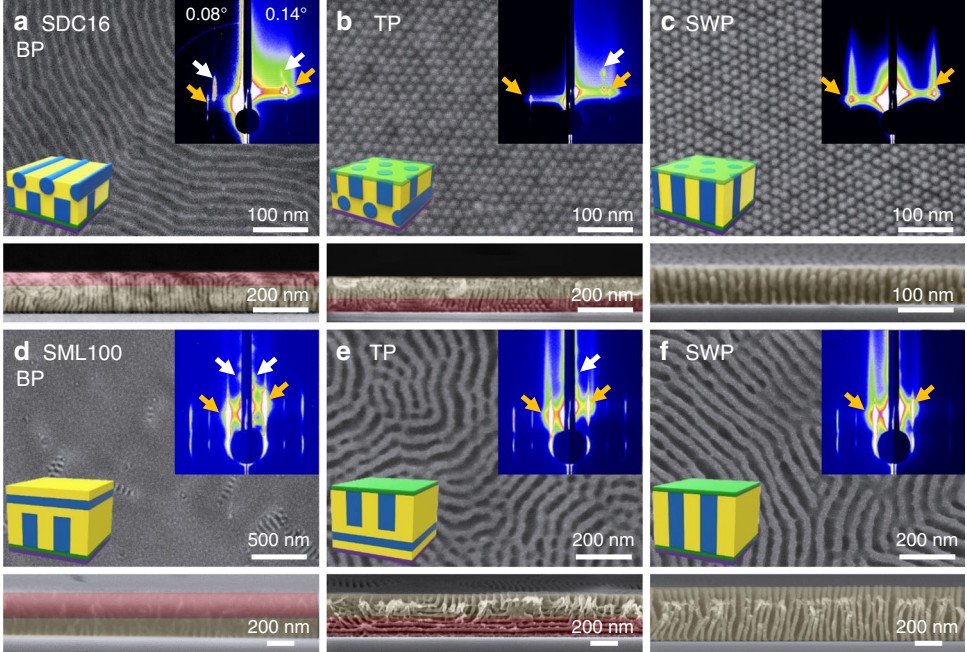

**Fig. 3** Orientation control of SDC16 and SML100 films with bottom, top and sandwich plasma-treated layers. Top-view and cross-section SEM images of **a–c** SDC16 and **d–f** SML100 with bottom plasma (BP, **a** and **d**), top plasma (TP, **b** and **e**) and sandwiched plasma (SWP, **c** and **f**). Inset schematics represent the microdomain orientations of BCP films with BP, TP, and SWP conditions. For BCP films with BP or TP, perpendicularly oriented BCP domains (yellow region) are observed at the interfaces in contact with the plasma treated layer, but parallel oriented BCP domains (red region) are observed at the untreated interface. In comparison, the BCP films with SWP condition have fully perpendicular orientation of microdomains throughout the film. Insets represent GISAXS patterns of the BCP films with incidence angles of 0.08° (left, below the critical angle $\alpha_c$) and 0.14° (right, above $\alpha_c$). The yellow and white arrows indicate the characteristic scattering peaks of the perpendicular or parallel orientations, respectively

coating ~7 nm of the BCP, applying the filtered plasma treatment, then spin-coating a second thicker layer of the BCP and annealing. Figure 3 shows top-view and cross-sectional morphologies of SDC16 (Fig. 3a–c) and SML100 (Fig. 3d–f) films under different surface and interface conditions. In the case of the bottom plasma-treated (BP) condition (Fig. 3a and d), perpendicular cylinder and lamellar structures originate from the bottom interfaces while parallel cylinders and lamellae are observed on the top surface. On the other hand, when only the top surface of the BCP film was plasma-treated (TP) (Fig. 3b, e), the cylinders and lamellae showed perpendicular orientation at the top surface but parallel orientation in the lower part of the film. When the plasma-treated layers were present at both the bottom and top interfaces (sandwich, SWP), the cylinder (Fig. 3c) and the lamellar (Fig. 3f) BCP structures formed a fully perpendicular orientation from the bottom to the top surface of the films. The fully-perpendicular oriented structures could be obtained at various film thicknesses from 25, 48 to 420 nm with the SWP condition, as can be seen in Supplementary Fig. 4.

Grazing incidence small-angle X-ray scattering (GISAXS) was used to investigate the microdomain orientation. By adjusting the angle of incidence below the critical angle, data is obtained from the surface of the film, whereas measurement above the critical angle probes the structure within the film. The critical angles for PS-*b*-PDMS and PS-*b*-PMMA are approximately 0.09°. Inset images in Fig. 3, Supplementary Figs. 5 and 6 show the GISAXS patterns at 0.08° and 0.14° incidence angles for SDC16 and SML100 films with different surface/interface conditions. The GISAXS data support the conclusions obtained from the SEM: TP-treated BCP films showed a perpendicular microdomain orientation at the top surface but parallel orientation inside the films, while the BP-treated films had parallel orientation on the surface but perpendicular orientation inside the films. The SWP BCP films exhibited perpendicular orientation both on the surface

and inside the film. A detailed analysis of the GISAXS patterns is described in the Supplementary Note 1.

**Comparison of characteristics according to presence of filter.** The effect of the filtered plasma on the etch resistance, chemical composition, and wetting angle of the BCP film was assessed. Figure 4a shows that films exposed to an unfiltered 200 W Ar plasma etched at 3 nm per second for PS-*b*-PMMA and ~2 nm per second for PS-*b*-PDMS, whereas the etch rate after exposure to a filtered plasma for 60 s was undetectable, i.e. the filter prevents chain scission induced by the VUV/UV light in the plasma. Angle resolved X-ray photoelectron spectroscopy (ARXPS), Fig. 4b, c, showed that 60 s filtered plasma treatment had no effect on the spectra of PS and PMMA homopolymers, whereas a 10 s unfiltered plasma treatment altered the C–C, C–O and aromatic peaks. The filtered plasma also had no effect on the water contact angles of the films, whereas the unfiltered plasma treatment lowered the contact angle, Fig. 4d–f. Details of these measurements are given in the Supplementary Note 2. They demonstrate that other than immobilizing the BCP, the filtered plasma has little or no effect on the chemical bonding and surface energy of the top surface of the film.

The effect of plasma treatment on the mechanical properties of the surface layer of the film was evident from examining the buckling instability of the films. A bilayer structure with large difference in elastic modulus between the layers undergoes a buckling instability when subjected to a compressive strain, causing micron-scale wrinkling. Such wrinkling is observed in polymer films exposed to ion bombardment which produces a high-density amorphous modified layer with high modulus and compressive stress[38,43]. In the SML100, SDC16, and SDC43 films, Ar plasma exposure led to micron-sized wrinkles in addition to the perpendicular microdomain orientation, but in the case of the filtered plasma, perpendicular orientation occurred without

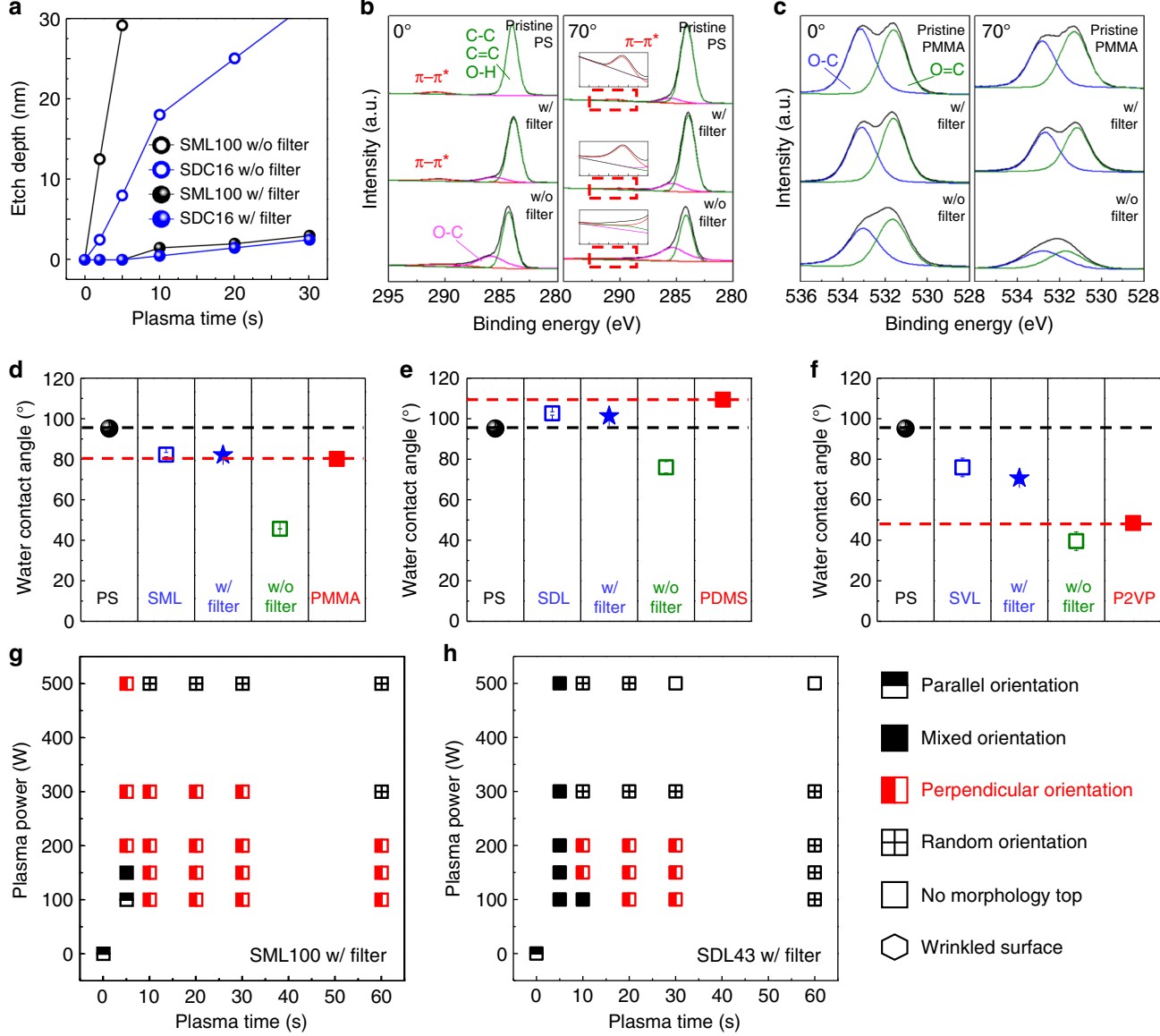

**Fig. 4** Surface characteristics of plasma treated polymer films with and without the filter. **a** Etch rate of PS-*b*-PMMA (SML100, black) and PS-*b*-PDMS (SDC16, blue) BCP films with (filled circle) and without (empty circle) the filter in a 200 W Ar plasma. **b**, **c** XPS spectra of PS (**b**, C1s) and PMMA (**c**, O1s) homopolymers: pristine film, after filtered plasma treatment (200 W, 60 s) and after plasma without filter (200 W, 10 s) with incidence angles of 0° and 70°. **d**-**f** Water contact angles of homopolymer films (black (PS) and red (PMMA, PDMS or P2VP, squares), as-cast BCPs (SML100, SDL43 or SVL84, blue hollow squares), filtered plasma treated BCPs (blue squares) and unfiltered plasma treated BCPs (green squares). Error bar represent the standard error of measured water contact angle from at least five independent experiments. **g**, **h** Orientational phase diagrams of (**g**) SML100 and (**h**) SDL43 treated with a filtered plasma for different powers and times. A red standing rectangle represents condition that yield perpendicular orientation of the microdomains, and a lying rectangle is a parallel orientation, a filled square is a mixed orientation, a crossed square is a partial perpendicular, an empty square is no morphology on top, and a hexagon is wrinkled morphology

wrinkling (Supplementary Fig. 7). This suggests that the limited modification of the polymer under the filtered plasma treatment produces insufficient modulus difference and compressive stress to drive buckling instability.

The duration and power of the filtered plasma determined the amount of cross-linking and therefore the alignment of the microdomains, as shown in Fig. 4g, h and Supplementary Figs. 8 and 9 for lamellar SML100 and SDL43 films. For SML100, a filtered Ar plasma treatment of less than 5 s produced a parallel terraced lamellar structure, increasing plasma energy and time promoted a mixed orientation, and a plasma of 150 W lasting 10 s or more led to a dominant perpendicular orientation, indicating that a threshold surface modification exists for forming a

perpendicular orientation. At higher energies, the cross-linked layer became thicker and denser, but these samples were still perpendicularly oriented and showed a flat surface as shown in Supplementary Fig. 10. These results indicate a wide process window for the filtered plasma treatment to ensure perpendicular orientation of the BCP after annealing. SDL43 treated with the filtered plasma also showed a process window for obtaining perpendicular microdomain orientation without wrinkling (Fig. 4h and Supplementary Fig. 7b). These results suggest that a process window for filtered plasma treatment can be determined for any BCP to ensure perpendicular orientation without wrinkling. In comparison, a process window also existed for the unfiltered plasma to produce perpendicular orientation of

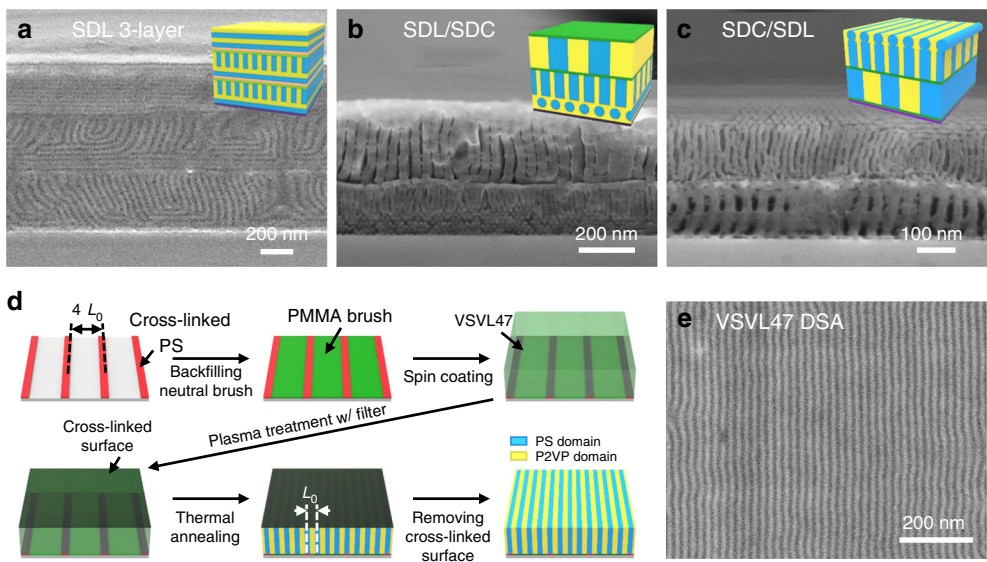

**Fig. 5** Filtered plasma method with 3D multilayer structures and DSA process. **a** SDL/P2VP/plasma/SDL/P2VP/plasma/SDL/substrate, **b** plasma/SDL/plasma/thin SDL/plasma/SDC/substrate and **c** SDC/plasma/thin SDC/plasma/SDL/plasma/thin SDL/substrate multilayers of PS-*b*-PDMS (SDL43 and SDC16) films with independent orientations can be fabricated using filtered plasma treatment for perpendicular orientation and P2VP thin layers (~7 nm) for parallel orientation. **d, e** P2VP-*b*-PS-*b*-P2VP (VSVL47) films with ~46 nm thickness on 84 nm period cross-linked PS/PMMA-OH stripes guide patterns for DSA. The films were treated by filtered plasma (200 W, 10 s) and thermally annealed at 240 °C for 3 h

the SML100 films, but all the films exhibited wrinkling (Supplementary Fig. 11).

**Multilayered 3D structures and DSA applications**. Our approach can be applied to create 3-dimensional multilayered structures with various orientations and to achieve DSA from chemoepitaxy. The former process is illustrated in Fig. 5a and Supplementary Fig. 12 which show a 3-layer SDL43 film (i.e., SDL/P2VP/plasma/SDL/P2VP/plasma/SDL/substrate) with perpendicular and parallel orientations forming a 90° angled lamellar structure. At each interface of the multilayer, the plasma treatment was applied to obtain perpendicular orientation in the underlying film, then a thin (~7 nm) P2VP interlayer was introduced by spin-coating from ethanol to obtain a parallel orientation for the subsequent BCP layer, allowing the orientation to be controlled independently for each layer. This demonstrates generation of arbitrary sequences of in plane and out of plane features by selection of plasma treatment, interlayer and BCP sequence. This method is also applicable to multilayers made from BCPs with different molecular weight or microdomain morphologies. In Fig. 5b, c, multilayers prepared with plasma/SDL/plasma/thin SDL/plasma/SDC/substrate and SDC/plasma/thin SDC/plasma/SDL/plasma/thin SDL/substrate respectively form 90° angled lamellae/cylinder structures with various orientations, determined by the interfacial properties at every interface. Similar to the finFET structure, Fig. 5c shows a hierarchical templated structure where one layer has vertical lamellae (fins), then in the next layer vertical cylinders emerge from the fins, and a further layer has horizontal cylinders connecting the tops of the vertical cylinders.

Our filtered plasma method is also compatible with DSA processes suitable for manufacturing. We prepared patterned substrates with line-space patterns consisting of cross-linked PS stripes with a thickness of 7.5 nm, a width of ~16 nm and a period of 84 nm (Supplementary Fig. 13), then attached a PMMA brush to the uncoated areas. This provides a chemical guide pattern for DSA of P2VP-*b*-PS-*b*-P2VP (VSVL47) because the PMMA brush has neutral affinity to PS and P2VP[44]. VSVL47 films with a thickness of 46 nm (~2 $L_0$) were spin-coated on the patterned

substrate, treated with a filtered plasma (Ar, 200 W, 10 s), and then thermally annealed at 240 °C for 3 h. Figure 5b shows that the plasma-treated VSVL47 film formed well aligned 9.5 nm-width line-space patterns. We also transferred the pattern into metal nanowires using selective deposition of metal salt in the P2VP microdomains[45] (Supplementary Fig. 14). This suggests that the location of the lamellae is guided by the substrate chemical pattern while the lithographically useful perpendicular orientation is maintained by the cross-linked top layer. Such orientation control provides one of the major check points for resolution enhancement of DSA process using high interaction parameter BCPs. The method using a conventional plasma etching tool is easily implemented and allows flexible selection of candidate BCPs for a sub-10 nm DSA process.

## Discussion

We show that a simple filtered Ar plasma treatment applied to the surface of a BCP film induces the perpendicular orientation of microdomains of various geometries, volume fractions, molecular weight, composition and annealing process. The filter blocks the ion bombardment and VUV/UV light generated within the plasma from reaching the film, which is subjected only to oblique collisions by neutral species. This leads to the formation of a thin cross-linked surface layer with the same chemical composition as the BCP, which provides a neutral surface to both blocks during subsequent annealing and promotes a perpendicular microdomain orientation. Filtering out the ion bombardment avoids formation of a dense carbonized surface layer as well as changes in the mechanical properties of the surface layer which can cause surface instability and wrinkling.

This method provides transformative capabilities in controlling the self-assembly of BCPs. It is ideally suited for obtaining perpendicular orientation of sub-10 nm microdomains from high interaction parameter BCPs because the difference in surface energy between the blocks generally increases with the block incompatibility. It can generate 3D structures such as lamellae or cylinders with 90° bends and junctions, and it is compatible with DSA to produce laterally-aligned sub-10 nm line-space patterns. We expect that the filtered plasma method will offer a universal

solution to the significant challenges of obtaining perpendicular orientation in self-assembled BCPs for next generation nanofabrication.

## Methods

**Plasma treatment of BCP film with the filter**. PS-*b*-PMMA (50 k(kg mol$^{-1}$)-48 k, SML100), PS-*b*-PDMS (11 k-5 k, SDC16 and 22 k-21 k, SDL43), PS-*b*-P2VP (40 k-44 k, SVL84), P2VP-*b*-PS-*b*-P2VP (12 k-23 k-12 k, VSVL47) and PMMA-*b*-PDMS (14 k-8 k, MDL22) from PolymerSource Inc. were used as purchased. For a top-plasma (TP) process, SML (30 mg ml$^{-1}$ in chloroform:acetone 9:1 mixture), SDC and SDL (30 mg ml$^{-1}$ in cyclohexane), SVL and VSVL (30 mg ml$^{-1}$ in chloroform) were spin-coated on the Si wafer at 6000 rpm to produce disordered BCP films with approximately 200 nm thickness. Then plasma treatment with a the filter was conducted using reactive ion etch equipment (PlasmaPro 800 RIE, Oxford Inc., 50 sccm Ar flow, 15 mTorr, different powers and times). The filter is placed on the sample stage of the RIE equipment to block normal-incidence photons and gas species. For a bottom-plasma (BP) condition (Supplementary Fig. 3), ~7 nm thick BCP films were prepared on Si wafers by spin-coating and exposed to Ar plasma with the filter (RF power 200 W, for 10 s). Then 200 nm thick BCP films are spin-coated on the plasma-treated thin BCP layers. The sandwiched plasma (SWP) conditions include the filtered Ar plasma treatment at both the top surface and the bottom of the films. The plasma treated BCP films were thermally annealed at 220 °C for 3 h in vacuum or solvent annealed with acetone for SDC16 and chloroform for SVL84 films in a 9.3 ml of chamber with a small leak for 30 min. The steady state swelling ratio of the films was ~1.5. The thickness changes of the plasma treated films were measured using spectral reflectometry (Filmetrics F20, USA).

**Characterization methods**. The samples were visualized using field emission scanning electron microscopy (FE-SEM, Zeiss Sigma 300, Zeiss) to observe the microdomain orientation in BCP films. An oxygen RIE was performed prior to SEM observation to remove the plasma treated surface layer and reveal the BCP microdomains. In the case of PS-*b*-PMMA, O$_2$ RIE (90 W, 10 sccm, 10 mTorr, 10 s) was sufficient to remove the surface layer and etch the PMMA to produce height contrast. In the case of PS-*b*-PDMS and PMMA-*b*-PDMS, two-step RIE was performed in which the surface layer was removed (CF$_4$ (10 sccm) + O$_2$ (10 sccm), 450 W, 20 mTorr, 15 s) and the PS domain was selectively etched (O$_2$ plasma, 90 W, 10 sccm, 10 mTorr, 10 s). For PS-*b*-P2VP and P2VP-*b*-PS-*b*-P2VP, after etching the surface layer (O$_2$ RIE 90 W, 10 sccm, 10 mTorr, 10 s), the films were exposed to iodine vapor in an isolated jar for 4 h to selectively stain the P2VP and enhance the contrast in the SEM observation.

Grazing Incidence Small-angle X-ray Scattering (GISAXS) measurements were made on the 3C and 9A beamline of the Pohang Light Source. A 2D detector consisting of 1920 × 1920 pixels was used to collect data. The X-ray wavelength was 0.6202 Å and the sample-to-detector distance was 3964 mm. The incident angle α was varied from 0.08° to 0.14°. The scattering results were analyzed using Fit2D software.

ARXPS were executed using a K-Alpha X-ray Photoelectron Spectrometer (Thermo Scientific Inc.). Survey scans were conducted at take-off angles of 0° and 70° to the surface normal to the sample and at different penetration depths. During the XPS analysis, the sample charge was compensated by a 200 mV e-beam at a high neutralization current by means of a flood gun. The pass energy was 200 eV for survey scans and 50 eV for high-resolution scans. The pressure during the analysis was kept below 7 × 10$^{-8}$ Torr. A 400 µm diameter beam was used in the analysis.

For a measurement of contact angles of the films, 200 nm thick films of PS, PMMA, P2VP homopolymers (Sigma-Aldrich) and, PS-*b*-PMMA (SML100), PS-*b*-PDMS (SDL43) and PS-*b*-P2VP (SVL84) were prepared by spin-coating on a Si wafer. PDMS homopolymer (Sylgard 184, Dow Corning Ltd.) film was spin-coated on Si wafer and cured at 180 °C for 6 h. The water contact angles were measured using Drop Shape Analyzer (DSA100, KRÜSS GmbH) immediately after plasma treatment with and without the filter (50 sccm of Ar flow, 15 mTorr, 200 W, for 10 s).

**Multilayered 3D structures and DSA applications**. For the multilayered 3D structures, 200 nm thick of SDL43 or SDC16 was first spin coated on a Si wafer. For a perpendicular orientation at the top surface, the filtered plasma process was executed. To obtain parallel orientation, a thin film (~7 nm) of P2VP was introduced on the TP-SDL43 films by spin-coating P2VP dissolved in ethanol. With repeated spin-coating and plasma treatment, SDL/P2VP/plasma/SDL/P2VP/plasma/SDL/substrate multilayer for Fig. 5a, plasma/SDL/plasma/thin SDL/plasma/SDC/substrate for Fig. 5b and SDC/plasma/thin SDC/plasma/SDL/plasma/thin SDL/substrate for Fig. 5c could be fabricated. The multilayer BCP films are thermally annealed at 220 °C for 3 h to produce multilayered 3D structures with diverse orientations.

For the DSA of BCPs with the plasma assisted perpendicular orientation, the cross-linked PS (xPS) guide pattern was prepared using a 300 mm fabrication process at imec. 13 nm thick SiN was first prepared as a BARC (bottom anti-reflective coating) layer by CVD. Then a 7.5 nm thick cross-linked PS (xPS) mat was prepared by spin coating of NLD-128 (Merck) and followed by a baking step (315 °C for 5 min). Using 193 immersion lithography, full pitch 84 nm line and

space pattern was printed on the xPS mat. N$_2$/O$_2$-based dry etch process removed the xPS exposed in the space areas while the width of line was trimmed down to the target value. A stripping process was performed using N-methylpyrrolidone (Sigma-Aldrich) to remove the remaining resist, resulting in xPS guide stripe (pitch 84 nm, width around 16 nm). The PMMA brush (hydroxyl terminated PMMA, 5k, PolymerSource Inc.) was grafted to the surface of the SiN/Si wafer between the PS stripes during an annealing process at 250 °C for 20 min and excess brush was rinsed with ultrasonic treatment in acetone (Sigma-Aldrich). On this chemical template, a VSVL47 film was spin coated with thickness around 46 nm. After filtered plasma treatment (200 W, for 10 s), the VSVL films (thickness = 2 $L_0$) were annealed at 240 °C for 3 h.

## Data availability
The data that support the findings of this study are available from the corresponding author upon request.

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

## Acknowledgements

We gratefully acknowledge financial support from the Korea Institute of Science and Technology (KIST) Institutional Program (Project No. 2E29520 and 2V06630) and the National Research Foundation of Korea (NRF) grant funded by the Korea government (MEST) (No. NRF-2019R1A2C2005657). C.A.R. acknowledges support from the Semiconductor Research Corporation. The experimental support by the staffs at the 3C and 9A beamlines of the Pohang Light Source is also gratefully acknowledged.

## Author contributions

J.O. and J.G.S conceived the idea and initiated the project. J.O. conducted the experiments. Y.N. prepared the samples with assistance from J.O. Y.K. conducted and analyzed the XPS characterization. H.S.S. prepared photolithographic patterned samples for DSA and discussed about the DSA with filtered plasma method. J.C.L., B.Y., and K.C. provided technical guidance. C.A.R. provided useful suggestions on the 3D multilayer designs and clear explanations of experiments. J.O., H.S.S., C.A.R., and J.G.S. co-wrote the manuscript. J.G.S. supervised the whole project. All authors commented on the manuscript.

## Additional information

**Competing interests:** The authors declare no competing interests.

