## [Peer Review File · Nature Communications]

Reviewers' comments:

Reviewer #1 (Remarks to the Author):

The manuscript by Oh et al details the results of experimental investigations of the morphology of block copolymer (BCP) thin films subjected to filtered (and non-filtered) plasma treatment before annealing. The principal finding here is that a relatively straightforward and quick (~ 10s) plasma treatment can be used to generate vertical orientation of the self-assembled structures produced by a variety of BCPs in ~ 200 nm thick BCP films. This manuscript continues the high quality contributions from a number of the senior investigators in this space.

The authors explore the parameter space associated with their process, and additionally demonstrate that the method, when combined with deposition of appropriate intervening layers, can be used to generate multilayer nanostructured thin films. The implications of this result are non-trivial, and the conclusions drawn follow quite directly from the data presented. From this perspective, I am supportive of publication. However the manuscript left a few important questions unanswered. Moreover, the manner in which the results are presented seems likely to steer readers to premature, incorrect conclusions on a number of important points. These issues are discussed below. A major revision seems appropriate.

1. The descriptions prior to p6 are misleading as they suggest that the morphology is one of vertical cylinders/lamellae throughout the film thickness; or at least there is no hint that the morphology produced by treatment of the top surface extends only a limited distance away from the top surface. The data of Figure 3 eventually make clear that the vertical ordering only persists for a certain distance away from the top crosslinked layer. I suggest the discussion of the results be conducted in a more straightforward manner, so that the reader gets the salient points quickly, and without drawing too many unnecessary false conclusions. I suggest also that a more complete description of the result of the plasma treatment be given in the abstract.

2. On p8 the authors indicate that vertical morphologies were produced with non-filtered plasma treatments (Figure S6), but the narrative until that point had suggested that filtering of the plasma was essential in some way. The drawback apparently in the non-filtered plasma treatment is that the films thus produced were subject to wrinkling during annealing. An earlier discussion on p8 asserted that the filtered plasma's principal effect was immobilization of the BCP surface layer. While it is reasonable to expect such an effect, this statement was made without corroborating data (or citation of prior literature) examining changes in the glass transition temperature (or other measures of chain mobility) in the near surface layer. This point should be addressed in some way, but a broader question however is why wrinkling was not observed in the filtered plasma treated samples, and whether there is a regime of process parameters for the non-filtered in which wrinkling could be avoided, and vice versa, for the filtered, in which wrinkling would be induced. Later, we see in Figure S10 that the authors are suggesting that at least for the parameter space considered, wrinkling could not be avoided for non-filtered plasma that produced perpendicular orientation. Then on p9 the discussion indirectly suggests that there *might* be a regime in which wrinkling could occur for filtered plasma treated films by asserting that a process window could be determined for any BCP to ensure perpendicular orientation *without wrinkling*. Some clarification is needed. It is also likely that a change in the manner of presenting the data would help to prevent readers from drawing incorrect conclusions prematurely (for example here, regarding the possibility of wrinkling from filtered plasma, and earlier, where a reader implicitly concludes that a non-filtered plasma may not produce perpendicular orientation, but later finds (Figure S10, and main text) that perpendicular orientations are possible, but apparently not without wrinkling).

3. The thickness of the films used in the study (~200 nm) are beyond the range that is typically considered for lithography applications. Did the authors explore thinner films, around 40-60 nm thickness? In the context of practical utility in lithography, some would argue that it's vitally important that the process be amenable to thinner films. Was there any challenge in producing continuous vertical orientation in the ~40 nm thickness regime, without excessive damage to the sample? Along this same line, was any pattern transfer attempted in these samples?

4. I am curious as to whether the method developed here could be applied to non PS-based BCPs. The selection of materials used in the study is somewhat understandable given the prevalence of PS-based materials in commercial libraries, and the general ease of synthesis/availability of well-developed synthetic routines for many PS-containing BCPs. That said, given the use of the word “universal” in the title of the manuscript, and the strong suggestion that the developed method should be applicable to a broad variety of polymers, I was struck by the commonality of PS across all the materials studied. Can the authors comment on this? Data showing the viability of the method for something like PB-PMMA, or PDMS-PnBA, or PDMS-PLA (all commercially available) would strengthen the claim.

MINOR

1. The text should make clear that the “sub-10 nm” terminology as used here apparently refers to feature size, and not system periodicity (L_0). On that note, PS-P4VP is a widely studied polymer (though not in the context of BCP lithography) that offers ready access to sub-10 nm periodicity nanostructures. Did the authors attempt their plasma treatment process with this (commercially available) BCP?
2. The data of Figure S10b shows data out to 60s of exposure time, and power levels up to 500 W (y-axis is incorrectly labelled with units of seconds). However the data in S10a, the morphology map, only covers a part of this parameter space. Is there a reason for this? I suggest S10a be expanded to adequately describe the full set of results obtained, as suggested by S10b.
3. On p5, I presume the word “radical” is referring to free radical chemistry. It may be helpful to insert the word “free” here to avoid ambiguity.

Reviewer #2 (Remarks to the Author):

The authors report of a new novel approach to improve block copolymer formation. The authors utilize a quite comprehensive literature from fields of plasma processing and materials characterization, which is impressive.

A few minor comments on the work are:

- 1) How long is the time between the filtered plasma exposure (low pressure) and the thick BCP coat (atmosphere)? Is there any dependency on the time between these steps?
- 2) The comment about this process being suitable for manufacturing on page 9 may be a bit exaggerated. The authors provide no mention of particle or contamination impact due to the filtered plasma approach. The authors also do not comment on the longevity of the polyamide coating and why it needs to be insulating.

Reviewer #3 (Remarks to the Author):

The authors present a universal methodology to achieve perpendicular orientation of micro domains in block copolymer thin films. Previous reports in the literature faced this problem providing partial solutions for specific classes of materials. In the manuscript the authors provide evidence that, in principle, the proposed approach can be extended to any kind of block copolymer, since the neutral layers that are necessary to promote perpendicular orientation can be created by proper modification of the material surface by UV filtered plasma exposure. I believe the manuscript presents an effective solution to the problem that could represent a breakthrough in the field. The data provided in the manuscript clearly demonstrate the inherent capabilities of the proposed approach. Systematic investigation of the effects of the plasma treatment are reported in the manuscript for 200 nm thick block copolymer films. Personally I believe that data about a set of samples with variable thickness could provide further details about the proposed methodology. In particular I wonder about the limit of the proposed methodology when scaling

down the thickness of the block copolymer film. Apart from this I believe the manuscript is a fantastic piece of work the definitely deserve publication on this journal.

Reviewers' comments:

Reviewer #1 (Remarks to the Author):

The manuscript by Oh et al details the results of experimental investigations of the morphology of block copolymer (BCP) thin films subjected to filtered (and non-filtered) plasma treatment before annealing. The principal finding here is that a relatively straightforward and quick (~ 10s) plasma treatment can be used to generate vertical orientation of the self-assembled structures produced by a variety of BCPs in ~ 200 nm thick BCP films. This manuscript continues the high quality contributions from a number of the senior investigators in this space.

The authors explore the parameter space associated with their process, and additionally demonstrate that the method, when combined with deposition of appropriate intervening layers, can be used to generate multilayer nanostructured thin films. The implications of this result are non-trivial, and the conclusions drawn follow quite directly from the data presented. From this perspective, I am supportive of publication. However the manuscript left a few important questions unanswered. Moreover, the manner in which the results are presented seems likely to steer readers to premature, incorrect conclusions on a number of important points. These issues are discussed below. A major revision seems appropriate.

1. The descriptions prior to p6 are misleading as they suggest that the morphology is one of vertical cylinders/lamellae throughout the film thickness; or at least there is no hint that the morphology produced by treatment of the top surface extends only a limited distance away from the top surface. The data of Figure 3 eventually make clear that the vertical ordering only persists for a certain distance away from the top crosslinked layer. I suggest the discussion of the results be conducted in a more straightforward manner, so that the reader gets the salient points quickly, and without drawing too many unnecessary false conclusions. I suggest also that a more complete description of the result of the plasma treatment be given in the abstract.

We really appreciate for your kind comment to improve our research results. As the reviewer commented, in the TP film, the perpendicular oriented morphology is observed limited distance from the top, however, the effective thickness of the top plasma treatments varies with the species, volume ratio and molecular weight of the BCP. We revised our description in page 5 of this part to, **“We demonstrate that the filtered plasma treatment produces a universal neutral top surface layer by spin-coating a BCP solution to form a disordered film, carrying out an Ar plasma treatment with the filter, then annealing the film to produce perpendicular orientation of the BCP microdomains at the top surface. To confirm the generality of our method, we successfully applied it to 200 nm thick films (sufficiently thick to minimize the effect of the bottom interface) of various block copolymers with different morphologies on pristine Si wafers.”**. And also, we added a phrase in the abstract **“This method induces the perpendicular orientation of microdomains of BCPs of various types, volume fractions and molecular weights. The orientation can be controlled not only from the top surface but also from the bottom interface by introducing a plasma-treated neutral layer underneath the BCP film.”** to clear the experimental results.

2. On p8 the authors indicate that vertical morphologies were produced with non-filtered plasma treatments (Figure S6), but the narrative until that point had suggested that filtering of the plasma was essential in some way. The drawback apparently in the non-filtered plasma treatment is that the films thus produced were subject to wrinkling during annealing. An earlier discussion on p8 asserted that the filtered plasma’s principal effect was immobilization of the BCP surface layer. While it is reasonable to expect such an effect, this statement was made without corroborating data (or citation of prior literature) examining changes in the glass transition temperature (or other measures of chain mobility) in the near surface layer. This point should be addressed in some way, but a broader question however is why wrinkling was not observed in the filtered plasma treated samples, and whether there is a regime of process parameters for the non-filtered in which wrinkling could be avoided, and vice versa, for the filtered, in which wrinkling would be induced.

Later, we see in Figure S10 that the authors are suggesting that at least for the parameter space considered, wrinkling could not be avoided for non-filtered plasma that produced perpendicular orientation. Then on p9 the discussion indirectly suggests that there *might* be a regime in which wrinkling could occur for filtered plasma treated films by asserting that a process window could be determined for any BCP to ensure perpendicular orientation *without wrinkling*. Some clarification is needed. It is also likely that a change in the manner of presenting the data would help to prevent readers from drawing incorrect conclusions prematurely (for example here, regarding the possibility of wrinkling from filtered plasma, and earlier, where a reader implicitly concludes that a non-filtered plasma may not produce perpendicular orientation, but later finds (Figure S10, and main text) that perpendicular orientations are possible, but apparently not without wrinkling).

We really appreciate for your concerns about the issues of the wrinkling phenomena under the plasma treatment with and without the filter. According to S. Tajima and K. Komvopoulos, Effect of reactive species on surface crosslinking of plasma-treated polymers investigated by surface force microscopy, *Applied Physics Letters*, **89**, 124102 (2006), plasma-induced surface cross-linking can be mechanically evaluated by frictional energy dissipated during nano-scratching. They used several shields during the Ar plasma to deconvolute the effect of various plasma species on the modification of the surface nanomechanical properties of polyethylene. The frictional energy of untreated polyethylene was 0.33 nJ, increasing greatly to 2.00 nJ after Ar plasma treatment due to the increase in the polymer shear resistance through the formation of a three dimensional network of cross-linked molecular chains. It was also found that the Ar plasma with an aluminum top shield passing only uncharged particles showed an increased friction energy of 1.10 nJ and therefore uncharged particles play an important role in surface cross-linking. Based on these results, we believe that our filtered plasma (which only allows collisions by uncharged particles) can cause a weak but adequate level of surface cross-linking.

For the wrinkling phenomena, we first refer to the paper, R. L. Bruce, G. S. Oehrlein *et al.* Relationship between nanoscale roughness and ion-damaged layer in argon plasma exposed polystyrene films *Journal of Applied Physics* **107**, 084310 (2010). This showed that Ar plasma treatment forms a dense amorphous carbon-like modified layer at the surface of a

wide range of polymers (PS, PMMA, etc.) with a thickness of a few nanometers. The modified layer densities obtained by ellipsometry are similar to densities measured for ion-enhanced amorphous carbon films. This high density and high elastic modulus is mainly from the "atomic peening" of ion bombardment. The ion bombardment causes atoms at the surface to be displaced, thereby creating a region of higher density and compressive stress. While carbon atoms become more densely packed, hydrogen was depleted from the surface, since displaced hydrogen atoms recombine to form H₂ gas.

It is clear then that plasma treatment forms a denser, stiffer surface layer on top of the unmodified polymer. Such a bilayer structure with large difference in elastic modulus between the layers undergoes a buckling instability which leads to wrinkle formation in the micron-scale range. Generally, for a thin, stiff film on a much thicker, softer substrate, a compressive strain applied in-plane to the stiff film above a critical value, ϵ_c , will create a buckling instability causing the bilayer structure to wrinkle. ϵ_c is

$$\epsilon_c = \left(\frac{3E_s}{8E_f} \right)^{2/3},$$

where E_f and E_s are the elastic modulus of the films and the substrate, respectively. This paper shows that there is an agreement between calculated values of wavelength and amplitude of wrinkling λ and A based on modified layer characteristics in conjunction with elastic buckling theory and measured values obtained by the analysis of the experimental wrinkled roughness patterns. This agreement strongly suggests that buckling caused by the ion-induced formation of a highly compressed, modified surface layer is the origin of the nanoscale roughness that developed during plasma etching. However, the paper said little modification by other plasma species (e. g. VUV, neutrals) is observed.

Based on the two papers, we conclude that our filtered plasma method forms sufficient cross-linking layer to provide a neutral layer that influences the microdomain orientation, but does not form an amorphous carbon-like layer from the atomic peening phenomena through the accelerated ion bombardment, so the modulus difference between the two layers and compressive stress were lower and buckling instability does not occur.

For consistency, we used the terms "collisions" for neutral particles and "bombardment" for ions during the plasma treatment in the manuscript. We also discussed the ion bombardment induced amorphous carbon-like modified layer in the manuscript, in page 4,

“During the plasma process, **accelerated Ar ions physically bombard the surface, forming a modified amorphous carbon-containing layer with a thickness of a few nanometers**³⁸,” and in page 5 “This physical collisions results in cross-linking of polymer chains only near the surface without chemical changes from the VUV/UV irradiation **or carbonization from accelerated ion bombardment**³⁸.”

Further, we rewrote the paragraph in page 8 about the wrinkling phenomena of filtered/non-filtered plasma, “The effect of plasma treatment on the mechanical properties of the surface layer of the film was evident from **the buckling instability of the films. A bilayer structure with large difference in elastic modulus between the layers undergoes a buckling instability when subjected to a compressive strain, causing micron-scale wrinkling. Such wrinkling is observed in polymer films exposed to ion bombardment which produces a high-density amorphous modified layer with high modulus and compressive stress.**^{38,43} **In the SML100, SDC16 and SDC43 films, Ar plasma exposure led to micron-sized wrinkles in addition to the perpendicular microdomain orientation, but in the case of the filtered plasma, perpendicular orientation occurred without wrinkling (Figure S7). This suggests that the limited modification of the polymer under the filtered plasma treatment produces insufficient modulus difference and compressive stress to drive buckling instability.**”

and in page 9, “**In comparison, a process window also existed for the unfiltered plasma to produce perpendicular orientation of the SML100 films, but all the films exhibited wrinkling.**”

3. The thickness of the films used in the study (~200 nm) are beyond the range that is typically considered for lithography applications. Did the authors explore thinner films, around 40-60 nm thickness? In the context of practical utility in lithography, some would argue that it's vitally important that the process be amenable to thinner films. Was there any challenge in producing continuous vertical orientation in the ~40 nm thickness regime, without excessive damage to the sample? Along this same line, was any pattern transfer attempted in these samples?

We appreciate to your comments about the thickness issues of the filtered plasma method for the BCP films. We prepared different thicknesses of the cylindrical PS-*b*-PDMS (SDC16) films from 25 nm to 420 nm and could obtain the perpendicular oriented cylinder structure with the filtered plasma method with sandwich-plasma condition. We added the sentence in the page 7, “**The fully-perpendicular oriented structures could be obtained at various film thicknesses from 25, 48 nm to 420 nm with the SWP condition, as can be seen in Figure S4.**” and we added SEM images of different thicknesses SDC16 films in **Figure S4**.

Figure S4. Cross-sectional SEM image of SDC16 with different film thickness of (a) 423 nm, (b) 182 nm, (c) 152 nm, (d) 67 nm, (e) 48 nm and (f) 25 nm.

Regarding pattern transfer: In Figure 5e, we also previously presented the 46 nm thick lamellar P2VP-*b*-PS-*b*-P2VP (VSVL) films for directed self-assembly on the pre-patterned substrate. We executed the pattern transfer process using the 46 nm VSVL films with selective deposition of Au salt to P2VP domains (refer to Jillian M. Buriak et al. *Nature Nanotechnology*, **2**, 500–506 (2007)). The P2VP lamellae was loaded with $[\text{AuCl}_4]^-$ by immersing the VSVL film into 5 mmol $\text{HAuCl}_4(\text{aq})$ / 1 mM HCl solution for 30 min. After exposing the film to oxygen plasma for 30 s, gold nanoparticles assembled in a line were observed. In figure S14, we added an SEM image of the transferred pattern of VSVL thin film. We referred to the pattern transfer in the manuscript on p10, “**We also transferred the pattern into metal nanowires using selective deposition of metal salt in the P2VP**

microdomains⁴⁵ (Figure S14).” and added in SI, “VSVL47 films with 46 nm thickness on 84 nm period xPS/PMMA stripes were used to exemplify pattern transfer. The cross-linked top surface of the VSVL47 film was removed by gentle oxygen plasma treatment (90 W for 5 s) and the film was immersed into 5 mmol H₂AuCl₄ / 1 mM HCl solution for 30 min. After removal of polymeric material with oxygen plasma treatment (150 W for 30 s), gold nanowires were obtained.” and added Figure S14.

Figure S14. SEM image of (a) I₂ stained VSVL and (b) pattern transferred gold nanowires after removing VSVL with O₂ plasma.

4. I am curious as to whether the method developed here could be applied to non PS-based BCPs. The selection of materials used in the study is somewhat understandable given the prevalence of PS-based materials in commercial libraries, and the general ease of synthesis/availability of well-developed synthetic routines for many PS-containing BCPs. That said, given the use of the word “universal” in the title of the manuscript, and the strong suggestion that the developed method should be applicable to a broad variety of polymers, I was struck by the commonality of PS across all the materials studied. Can the authors comment on this? Data showing the viability of the method for something like PB-PMMA, or PDMS-PnBA, or PDMS-PLA (all commercially available) would strengthen the claim.

We really appreciate for the kind comment to broaden the variety of polymers. A lamellar PMMA-*b*-PDMS diblock (14 kg/mol – 8 kg/mol, MDL22, $f_{PS} = 0.58$) copolymer were used

to strengthen our claims. The MDL22 film annealed with thermal treatment shows a parallel lamellar structure. This shows that our approach works for a non-PS-containing block copolymer, MDL22, and perpendicularly oriented lamellae were observed with SEM. These data were added to **Figure 2d (thermal annealing)**. Thank you again about the great comment for our experiments.

Figure 2. Morphology differences of various annealed BCP films with and without the filtered plasma treatment. Top view SEM images of thermally annealed PS-*b*-PMMA (**a**, SML100), lamellar PS-*b*-PDMS (**b**, SDL43), P2VP-*b*-PS-*b*-P2VP (**c**, VSVL47), **PMMA-*b*-PDMS** (**d**, MDL22) and cylinder PS-*b*-PDMS (**e**, SDC16), and solvent annealed (**f**) SDC16 and (**g**) PS-*b*-P2VP (SVL84) without (left) and with (right) the filtered plasma treatment. The annealed BCP films without the plasma treatment showed (**a-d** and **g**, left) a terraced structure from incommensurate thickness of the parallel lamellae and (**e** and **f**, left) parallel cylindrical microdomains. On the other hand, perpendicular orientation of lamellae (**a-d** and **g**, right) and cylinders (**e** and **f**, right) were obtained on the surface of various BCP films with filtered plasma treatment.

MINOR

1. The text should make clear that the “sub-10 nm” terminology as used here apparently refers to feature size, and not system periodicity (L₀). On that note, PS-P4VP is a widely studied polymer (though not in the context of BCP lithography) that offers ready access to sub-10 nm periodicity nanostructures. Did the authors attempt

their plasma treatment process with this (commercially available) BCP?

Thank you for the good suggestion about the PS-*b*-P4VP. We tried to execute the self-assembly of 2 different PS-*b*-P4VP (Figure R1 (a) 27k – 17k and (b) 36.5k – 16k) in thin film geometry. However, without any plasma treatment, thermally annealed PS-*b*-P4VP films (at 180 °C, for 16 hr) did not show any parallel oriented BCP microdomains and rather exhibited micellar morphologies. Spherical micelle formation is a common observation in PS-*b*-P4VP (e.g. Park et al, *ACS Nano*, 2008, 2 (7), pp 1363–1370). Because we could not obtain in-plane morphologies even with high volume fractions of P4VP, we could not test our filtered plasma method to control the orientation of PS-*b*-P4VP films.

Figure R1. SEM images of 200 nm thick (a) 27k – 17k and (b) 36.5k – 16k of PS-*b*-P4VP thin films after thermal annealing at 180 °C for 16 hr in vacuum.

2. The data of Figure S10b shows data out to 60s of exposure time, and power levels up to 500 W (y-axis is incorrectly labelled with units of seconds). However the data in S10a, the morphology map, only covers a part of this parameter space. Is there a reason for this? I suggest S10a be expanded to adequately describe the full set of results obtained, as suggested by S10b.

Thank you for your kind comment. We mistyped in the morphology and SEM maps. The power level and the morphology map were corrected in Figure S10b.

3. On p5, I presume the word “radical” is referring to free radical chemistry. It may be helpful to insert the word “free” here to avoid ambiguity.

We really appreciate for your meticulous comment about the terminology “radical”. We accepted your comment and changed the term from “radical” to “free radical”.

Reviewer #2 (Remarks to the Author):

The authors report of a new novel approach to improve block copolymer formation. The authors utilize a quite comprehensive literature from fields of plasma processing and materials characterization, which is impressive.

A few minor comments on the work are:

1) How long is the time between the filtered plasma exposure (low pressure) and the thick BCP coat (atmosphere)? Is there any dependency on the time between these steps?

Thank you for the question about the time dependency of the neutrality of the plasma treated BCP layer. The time gap between two processes appears not to be important even we leave the sample in atmosphere for a day. The time gap was varied from tens of minutes to several hours. In most cases, the thick BCP layer was coated **30 min to 1 hour** after plasma exposure. Because our process does not change the hydrophilicity/phobicity of the film surface (from the contact angle result in Figure 4d-f), we think that surface recovery from hydrophilic to hydrophobic which generally occurs in plasma treated polymers is not present in our system. The filtered plasma treated layer even works as a neutral layer during the thermal annealing at 220 °C.

2) The comment about this process being suitable for manufacturing on page 9 may be a bit exaggerated. The authors provide no mention of particle or contamination impact due to the filtered plasma approach. The authors also do not comment on the longevity of the polyamide coating and why it needs to be insulating.

Thank you very much for your concerns about our filtered plasma system. In conventional plasma processing processes, gases or particles (dust) from previous processes or from a contaminated chamber can cause unwanted effects on the film surface, even if the chamber is cleaned regularly. In the result of XPS for the Ar plasma treated BCP film **without filter**, fluorine and chloride not used in the experiment were detected at the film surface in Figure S15. This likely originated from the plasma reactor. In contrast, the Ar plasma treated BCP

film *with the filter* is free of chemical contamination. The filter blocks not only UV but also accelerated ions which can attach to the film surface. We added a comment about the chemical contamination during the plasma at the part of details of ARXPS in the supplementary information, “In the full-scan XPS spectra for the PS film after Ar plasma treatment without the filter in Figure S15, fluorine and chloride not used in the experiment were detected at the film surface. This contamination may originate from previous use of these materials in the reactor and is undesirable. In contrast, the PS film after Ar plasma treatment with the filter did not exhibit any chemical contamination because the filter blocks not only UV but also accelerated ions from striking the film surface.” and **Figure S15**.

Figure S15. XPS data of Ar plasma treated polystyrene w/ (red) and w/o filter (black).

The filter was prepared with SUS. During the plasma treatment, charged particles such as electrons and ions are accelerated by the electric field generated during the RIE, and the conductive plate can change the electric field arrangement. To minimize this, a polyimide (PI) coating was performed to separate the SUS filter from the plasma chamber. However, the SUS mask without PI coating can also induce some cross-linking on the top surface and the perpendicularly oriented cylinder and lamellae microdomain of PS-*b*-PDMS and PS-*b*-PMMA was equally obtained after filtered plasma with SUS filter, which can be seen in Figure R2. We additionally comment in the figure caption of Figure S1, “Masks without polyimide coating also showed the same results.”

Figure R2. SEM image of SDC16(a, d), SDL43(b, e) and SML100(c,f) after plasma treatment with SUS filter (a-c) and PI coated SUS filter (d-e). Cylinder and lamellae microdomains with perpendicular orientation are observed with plasma treatment regardless of the filter material.

Reviewer #3 (Remarks to the Author):

The authors present a universal methodology to achieve perpendicular orientation of micro domains in block copolymer thin films. Previous reports in the literature faced this problem providing partial solutions for specific classes of materials. In the manuscript the authors provide evidence that, in principle, the proposed approach can be extended to any kind of block copolymer, since the neutral layers that are necessary to promote perpendicular orientation can be created by proper modification of the material surface by UV filtered plasma exposure. I believe the manuscript presents an effective solution to the problem that could represent a breakthrough in the field. The data provided in the manuscript clearly demonstrate the inherent capabilities of the proposed approach. Systematic investigation of the effects of the plasma treatment are reported in the manuscript for 200 nm thick block copolymer films. Personally I believe that data about a set of samples with variable thickness could provide further details about the proposed methodology. In particular I wonder about the limit of the proposed

methodology when scaling down the thickness of the block copolymer film. Apart from this I believe the manuscript is a fantastic piece of work the definitely deserve publication on this journal.

We really appreciate about your kind comment. We controlled the film thickness of the cylindrical PS-*b*-PDMS (SDC16) from the thick films (~420 nm) to thinner films (25, 48 and 67 nm). The all BCP films with sandwiched filtered plasma show fully-perpendicularly oriented cylindrical structure. We added the sentence in the page 7, “**The fully-perpendicular oriented structures could be obtained at various film thicknesses from 25, 48 nm to 420 nm with the SWP condition, as can be seen in Figure S4.**” and SEM images of different thicknesses SDC16 films in **Figure S4**. Moreover, thin lamellae P2VP-*b*-PS-*b*-P2VP film with 46 nm thick also showed vertically oriented lamellae structures when we executed the DSA process with the underling chemical patterns (**Figure 5e**). Thus, we think our method can be fully compatible with the lithographic applications.

Figure S4. Cross-sectional SEM image of SDC16 with different film thickness of (a) 423 nm, (b) 182 nm, (c) 152 nm, (d) 67 nm, (e) 48 nm and (f) 25 nm.

Figure 5e. SEM image of 46 nm thick VSVL films (I₂ stained) with filtered plasma on the pre-patterned substrate for DSA process.

REVIEWERS' COMMENTS:

Reviewer #1 (Remarks to the Author):

The authors have provided responses and performed revisions that suitably address the concerns raised during the prior round of review. The additional data re: thin films, and work on non PS-containing BCPs was particularly welcome. Publication in the current form is recommended.

Reviewer #2 (Remarks to the Author):

The Authors have adequately addressed the questions raised by the reviewers.

Reviewer #3 (Remarks to the Author):

The manuscript has been improved compared to the previous version introducing new data about the application of the proposed methodology to polymeric films with thickness ranging from 25 to 420 nm. Moreover the authors introduced new BCP materials and demonstrated the possibility of pattern transfer to the underlying substrate. I confirm my previous evaluation that the present manuscript is an excellent piece of work and definitely deserve to be published.

Reviewers' comments:

Reviewer #1 (Remarks to the Author):

The authors have provided responses and performed revisions that suitably address the concerns raised during the prior round of review. The additional data re: thin films, and work on non PS-containing BCPs was particularly welcome. Publication in the current form is recommended.

We really appreciate for your kind comment to improve our research results and recommend to publish.

Reviewer #2 (Remarks to the Author):

The Authors have adequately addressed the questions raised by the reviewers.

Thank you very much for commenting us that we adequately addressed.

Reviewer #3 (Remarks to the Author):

The manuscript has been improved compared to the previous version introducing new data about the application of the proposed methodology to polymeric films with thickness ranging from 25 to 420 nm. Moreover the authors introduced new BCP materials and demonstrated the possibility of pattern transfer to the underlying substrate. I confirm my previous evaluation that the present manuscript is an excellent piece of work and definitely deserve to be published.

We really appreciate about your kind evaluation that the manuscript is an excellent work and definitely deserve to be published.